# The Complex Dynamic of Phase I Drug Metabolism in the Early Stages of Doxorubicin Resistance in Breast Cancer Cells

**DOI:** 10.3390/genes13111977

**Published:** 2022-10-29

**Authors:** Isabel S. Barata, Bruno C. Gomes, António S. Rodrigues, José Rueff, Michel Kranendonk, Francisco Esteves

**Affiliations:** Centre for Toxicogenomics and Human Health (ToxOmics), Faculdade de Ciências Médicas (FCM), NOVA Medical School (NMS), Universidade Nova de Lisboa, 1169-056 Lisboa, Portugal

**Keywords:** breast cancer (BC), drug metabolism enzymes (DMEs), drug resistance (DR), doxorubicin (DOX), cytochrome P450 (CYP), oxidoreductases

## Abstract

The altered activity of drug metabolism enzymes (DMEs) is a hallmark of chemotherapy resistance. Cytochrome P450s (CYPs), mainly CYP3A4, and several oxidoreductases are responsible for Phase I metabolism of doxorubicin (DOX), an anthracycline widely used in breast cancer (BC) treatment. This study aimed to investigate the role of Phase I DMEs involved in the first stages of acquisition of DOX-resistance in BC cells. For this purpose, the expression of 92 DME genes and specific CYP-complex enzymes activities were assessed in either sensitive (MCF-7 parental cells; MCF-7/DOX^S^) or DOX-resistant (MCF-7/DOX^R^) cells. The DMEs genes detected to be significantly differentially expressed in MCF-7/DOX^R^ cells (12 CYPs and eight oxidoreductases) were indicated previously to be involved in tumor progression and/or chemotherapy response. The analysis of CYP-mediated activities suggests a putative enhanced CYP3A4-dependent metabolism in MCF-7/DOX^R^ cells. A discrepancy was observed between CYP-enzyme activities and their corresponding levels of mRNA transcripts. This is indicative that the phenotype of DMEs is not linearly correlated with transcription induction responses, confirming the multifactorial complexity of this mechanism. Our results pinpoint the potential role of specific CYPs and oxidoreductases involved in the metabolism of drugs, retinoic and arachidonic acids, in the mechanisms of chemo-resistance to DOX and carcinogenesis of BC.

## 1. Introduction

Breast cancer (BC) is the second most common cancer diagnosed in women and the leading cause of death from cancer in women worldwide [1]. Chemotherapy is one of the main approaches in the treatment of BC. However, a lack of efficacy due to intrinsic or acquired drug resistance (DR) is a major impediment in chemotherapy, resulting in increased disease progression, relapses, and, eventually, death [1,2,3,4,5,6]. Acquired DR is developed during therapy, resulting from complex selective and adaptive processes, including alterations in drug transport and metabolism (increased efflux, decreased uptake, enhanced detoxification), in drug targets, and/or programmed cell death inhibition [5,6,7,8,9,10,11,12]. However, the majority of molecular mechanisms leading to these compromising variations in drug response remain largely unexplained.

Although variability of drug metabolism in the liver (main site of drug metabolism) must be considered as a potential factor mediating drug sensitivity or resistance, intra-tumoral expression of drug metabolism enzymes (DMEs), including cytochrome P450s (CYPs), plays an important role in regulating the efficacy of drugs [4,13,14]. DMEs expression in BC cells significantly affects drug response and the onset of resistance to therapy by accelerating the degradation and clearance of anti-cancer agents in tumor cells.

Doxorubicin (DOX) is an anthracycline commonly used as a chemotherapeutic agent in BC treatment. It is typically administered in combination with other chemotherapy medications, at a maximum plasma concentration of 6.7 µM [1,2,15]. DOX is a topoisomerase II inhibitor and generates free radical-mediated oxidative damage to DNA, inducing apoptosis. Oxidation mediated by CYPs, particularly CYP3A4 (and CYP2D6, 2B6, 1B1 to a minor extent), is considered the main primary metabolic pathway of DOX metabolism, while the one-electron reduction and deglycosidation, both facilitated by oxidoreductases (e.g., CYP-reductase (CPR), NADH- and NADPH dehydrogenases, xanthine dehydrogenase), are considered secondary minor routes [16,17,18]. Efflux of DOX is dependent on several members of the ABC transporters family (including ABCB1, ABCC1, and ABCG2), and solute carrier family (e.g., SLC22A16) [2,8,18].

Efflux transporters, particularly members of the ABC family, have been widely studied in acquired DOX-resistance in BC cells. Several studies, both in vivo and in vitro, established the relationship between DOX-resistance and the overexpression of ABCB1, identified as the main DOX efflux transporter [2,5,6,8,19]. A previous study using a human chronic myeloid leukemia-derived cell line resistant to imatinib and dasatinib, demonstrated that the mRNA expression pattern of efflux transporters varies over time with resistance level and chronic drug exposure, suggesting that other mechanisms are also dynamically involved in DR [3,9]. Due to their involvement in the metabolism of both endogenous and exogenous substances, Phase I DMEs are crucial in terms of tumor development and response to therapy [2,18]. However, there is a scarcity of data regarding the role of Phase I DMEs, including CYP-complex enzymes and other oxidoreductases, in the mechanisms of DOX-resistance. This is of particular importance in the initial stages of resistance acquirement, i.e., at low levels of DOX, which may instigate/enable the formation of DR at therapeutic concentrations.

CYPs, together with several oxidoreductases, are involved in the metabolism of xenobiotics, sterols, fatty acids, eicosanoids and vitamins [7,20,21,22]. CYP isoforms of the families 1–3 are key Phase I microsomal enzymes in the biotransformation of a wide range of anti-cancer drugs (including DOX), which are metabolized primarily in the liver and additionally in tumor tissues [2,16,17,20]. Although sporadically studied, deregulation of the expression and/or activity of these CYPs has been suggested to be involved in chemotherapy failure [7,23,24]. Regarding tumorigenesis, mitochondrial CYPs, involved in sterol and vitamin metabolism, seem to be deregulated in BC cells [22,25,26]. Recently, the expression and activity profiles of several CYPs and oxidoreductases have been investigated as putative tumor biomarkers. The association between expression of these enzymes and cancer risk, tumorigenesis, progression, metastasis, and prognosis has been widely reported in basic, clinical and epidemiological studies [23,27,28,29,30,31,32,33,34]. Nevertheless, their role has not been properly established in the formation of DR.

In order to achieve this, we investigated how Phase I DMEs contributed to the early development of DOX resistance in BC cells. To understand their significance in the early phases of DR, we examined and analyzed the expression profiles of 92 genes, including most representative CYP isoforms and Phase I oxidoreductases, in a BC luminal A cell line (MCF-7), which was either sensitive or resistant to low doses of DOX. Measurements of pertinent CYP-enzyme activities were added to the study in order to assess the effects of changing transcript levels.

## 2. Materials and Methods

### 2.1. Reagents

Cytochrome *c* (cyt *c*) (horse heart), glucose-6-phosphate (G6P), glucose-6-phosphate dehydrogenase (G6PD), nicotinamide-adenine dinucleotide phosphate (NADPH), ethoxyresorufin, methoxyresorufin, resorufin, coumarin, 7-hydroxy coumarin, 3-cyano-7-ethoxycoumarin, 3-cyano-7-hydroxycoumarin, fluorescein, trypsin, penicillin-streptomycin (10,000 units penicillin and 10 mg streptomycin per mL), Dulbecco’s Modified Eagle’s Medium-low glucose (DMEM), fetal bovine serum (FBS), phosphate buffered saline pH 7.4 (PBS) and doxorubicin were obtained from Sigma Aldrich (St. Louis, MO, USA). Nicotinamide adenine dinucleotide phosphate (NADP+) was obtained from Gerbu (Heidelberg, Germany). Bradford reagent was obtained from Bio-Rad (Hercules, CA, USA) and Quiazol from Qiagen (Hilden, Germany). Dibenzylfluorescein, sodium dithionite, dimethyl sulfoxide (DMSO), acetonitrile (ACN) and sodium hydrogencarbonate (NaHCO_3_) were purchased from Merck (Kenilworth, NJ, USA). Insulin was obtained from Cell Applications, Inc. (San Diego, CA, USA). All other chemicals and solvents were of the highest grade commercially available.

### 2.2. MCF-7 Cells Cultures

The MCF-7 cell line, a human breast adenocarcinoma cell line with luminal A subtype and naturally sensitive to DOX, was purchased from DSMZ-German Collection of Microorganisms and Cell Culture GmbH (Braunschweig, Germany) (MCF-7, ACC 115). DOX-resistant (DOX^R^) cells were engineered by stepwise exposures to increasing concentrations of DOX. Cells were cultured in DMEM medium supplemented with 10% FBS, 1% penicillin-streptomycin and 10 μg/mL insulin, in a 5% CO_2_ incubator at 37 °C. Cells resistant to 25 or 35 nM DOX (MCF-7/DOX^R^) were obtained by supplementing the culture medium with DOX in incremented doses according to cell response. An untreated DOX-sensitive parental control (MCF-7/DOX^S^) was cultured in parallel.

### 2.3. Evaluation of mRNA Expression Levels of CYPs and Oxidoreductases in MCF-7/DOX^R^ Cells

#### 2.3.1. RNA Isolation and cDNA Synthesis

RNA was isolated from MCF-7/DOX^R^ 25 nM, MCF-7/DOX^R^ 35 nM and MCF-7/DOX^S^ cells. Approximately 3 × 10^6^ cells were washed with cold PBS buffer and centrifuged at 100× *g* for 5 min. The cell pellet was resuspended in 700 μL of Quiazol and frozen at −80 °C for later use, following the protocol provided with the Direct-zol™ RNA Miniprep Plus kit (Zymo Research, Irvine, CA, USA). cDNA was prepared from the total RNA isolated, using the High-Capacity RNA-to-cDNA™ Kit (Applied Biosystems, Waltham, MA, USA), with 1.7 µg of total RNA per 20 µL reaction, according to the manufacturer’s instructions. The cDNA synthesized was stored at −20 °C until use.

#### 2.3.2. RT-qPCR

The expression profile of 92 genes (Appendix A) was quantified (duplicate) in MCF-7/DOX^R^ 25 nM, MCF-7/DOX^R^ 35 nM and MCF-7/DOX^S^ cells, using the TaqMan™ Array Human CYP450 and other Oxygenases 96-Well Plates (Applied Biosystems, Waltham, MA, USA) in a QuantStudio 5 Real Time PCR system (Applied Biosystems, Waltham, MA, USA), with 47.2 ng of cDNA per well, following the manufacturer’s protocol. The mean values of the duplicate RT-qPCR reactions for each gene expression assay were normalized using three endogenous controls (GADPH, HPRT1 and GUSB). The relative expression (fold-change) of the target genes was determined by the 2^–∆∆Ct^ method.

### 2.4. Analysis of CYP-Mediated Activities in MCF-7/DOX^R^ Cells-Derived Microsomes

#### 2.4.1. Subcellular Fractions (Protein/Microsomes) Isolation and Characterization

Membrane proteins were isolated from MCF-7/DOX^S^ and/DOX^R^ cells using the Mem-PER Plus Membrane Protein Extraction Kit (Thermo Scientific, Waltham, MA, USA), following the manufacturer’s instructions for membrane protein extraction from mammalian cells. Briefly, cells were trypsinized, harvested by centrifugation at 100× *g* for 5 min and resuspended in growth media. Approximately 5 × 10^6^ cells were washed with the cell wash solution provided with the kit, centrifuged at 300× *g* for 5 min and the supernatant discarded. After the addition of the permeabilization buffer, supplemented with a protease inhibitor cocktail (#11 836 153 001, Roche, Basel, Switzerland), the samples were briefly vortexed and incubated for 10 min at 4 °C, with constant mixing. The suspension obtained was centrifuged for 15 min at 16,000× *g* and the supernatant discarded; the pellet was resuspended in solubilization buffer and incubated for 30 min at 4 °C with constant mixing. After centrifugation, the supernatant (containing solubilized membrane and membrane-associated proteins) was stored at −80 °C, until used.

Total membrane proteins were quantified using the Bradford method [35]. CYP quantification was performed by CO-difference spectrophotometry and CPR by cytochrome *c* reduction, similarly to what was previously described [36,37,38]. Due to the low CYP concentration in BC cell lines, a membrane protein fraction isolated as previously reported [39] from an in-house engineered bacterial cell model co-expressing CPR and CYP1A2 [40,41] was added to the MCF-7 cells derived membrane protein fractions in order to increase the signal-to-noise ratio. Briefly, 100 µL of this bacterial membrane protein fraction was added to 50 µL of the MCF-7 membrane protein fractions, 1650 µL of cold TGE buffer (75 mM Tris, 25 mM EDTA, 10% glycerol, pH 7.5) and 15 µL of 100 mg/mL sodium dithionite; for the control, buffer was used instead of the MCF-7 membrane protein fractions. The CO-difference spectra were traced between 400 and 500 nm and CYP concentration determined as previously reported [36,42].

#### 2.4.2. CYP-Activity Assays

Specific CYP-mediated activity assays were performed with microsomal fractions isolated from MCF-7/DOX^R^ 25 nM, MCF-7/DOX^R^ 35 nM and MCF-7/DOX^S^ cells. These measurements were performed using different standard probe substrates reactions (MROD: methoxyresorufin O-demethylation; EROD: ethoxyresorufin O-deethylation; CECOD: cyanoethoxycoumarin O-dealkylation; C7H: Coumarin 7-hydroxylation; DBFOD: dibenzylfluorescein O-debenzylation), as previously described [39,43]. The substrate concentrations selected (1 µM MROD, 2.5 µM EROD, 25 µM CECOD, 5 µM C7H and 3.75 µM DBFOD) were above the K_M_ values determined in previous studies: 0.59 µM, 1.16 µM and 5.00 µM, respectively, for MROD, EROD and CECOD CYP1A2-mediated activities [44], 1.57 µM for 2A6-mediated C7H, and 0.89 µM for 3A4-mediated DBFOD [43]. The assays were conducted in triplicate with a final MCF-7 microsomes total protein concentration of 0.2 mg/mL per well.

### 2.5. Statistical Analysis

Regarding RT-qPCR data, a mixed-effects model (REML) with Dunnett’s multiple comparisons test was used to test statistical significance and determine the *p*-values; differential gene expression was considered when obtaining *p* < 0.05 and fold change < 2 or >2. Unpaired *t* tests were performed to compare protein levels and relative velocities of the CYP enzyme assays. Data was analyzed using GraphPad Prism 8.4.3 software (La Jolla, CA, USA).

## 3. Results

### 3.1. Gene Expression Profiles of CYPs and Oxidoreductases in the Initial Stages of DOX-Resistance in MCF-7 Cells

The expression of CYP-enzyme complex protein factors and oxidoreductases in conditions of low levels of resistance to DOX, was assessed by profiling the expression of 92 target genes from MCF-7 cell lines, either sensitive or engineered to be resistant to sub-therapeutic concentrations of DOX. The fold change and respective *p*-values were used to determine the differential scores of mRNA levels in the two types of MCF-7/DOX^R^ cells versus the MCF-7/DOX^S^ cells. From the 92 target genes evaluated, 20 (12 CYPs and 8 oxidoreductases) were found to be differentially expressed in the initial stages of DOX-resistance, when compared to the MCF-7/DOX^S^ cells (Figure 1). From these, the majority were overexpressed, with three exceptions, namely *CYP2D6*, *2S1* and *3A5* which were downregulated. Interestingly, the expression of eight genes was found to be deregulated in MCF-7/DOX^R^ 25 nM cells, which deregulation (up or down) was found amplified (doubled) in MCF-7/DOX^R^ 35 nM cells. Only five genes (*CYP4F12*, *8B1*, *26B1*, kynurenine 3-hydroxylase (*KMO*), phenylalanine hydroxylase (*PAH*)) were overexpressed in cells resistant to both DOX levels. This consistency is indicative that expression of these five genes is key in the mechanisms of initial DOX-resistance of MCF-7 cells, when exposed to increasing concentrations of DOX (up to 35 nM). Three genes (*CYP2A6*, *2D6*, *2S1*) seem to be transiently involved in early cell response to selective pressure, as they were found to be differentially expressed only in the MCF-7/DOX^R^ 25 nM cells. Additionally, CYP1A2 mRNA was detected only in the MCF-7/DOX^S^ cells, evidencing a putative downregulation of *CYP1A2* expression in MCF-7/DOX^R^ cells.

Transcripts of the CYP11 family, as well as *CYP2A13*, *CYP2F1*, *CYP7B1*, prostaglandin I2 synthase (*PTGIS*), and dopamine β-hydroxylase (*DBH*), were not amplified from any of the RNA samples obtained from either sensitive or resistant cells. This is indicative that these transcripts were present in concentrations below the detection threshold of the RT-qPCR. This might be the result of very low, or even inexistent, levels of transcripts of these genes.

### 3.2. Detailed CYP-Dependent Activities in MCF-7/DOX^R^ Cells-Derived Microsomes

Microsomal fractions were isolated from MCF-7/DOX^S^ and MCF-7/DOX^R^ 25 nM and characterized for CYP-mediated activities. The microsomal contents of components of the CYP-enzyme complex system are shown in Table 1. Microsomal CYP isoforms are strictly dependent on CPR in their activity [21,22,37]. In vivo, CPR:CYP contents are in favor of CYP, implying competition between individual CYP isoforms in binding to CPR in the endoplasmic reticulum [45,46]. Although no significant differences in total CYP contents were observed, both MCF-7 microsomal fractions evaluated (MCF-7/DOX^S^ and MCF-7/DOX^R^ 25 nM) have lower total CYP contents, when compared with human liver microsomes (ranging from 210 to 580 pmol/mg) [46,47]. CPR content was significantly higher (1.4 × fold; *p* < 0.005) in microsomes of MCF-7/DOX^R^ 25 nM cells. In addition, the CPR:CYP ratios determined in the MCF-7 microsomal fractions were found to be higher than those reported previously for human liver microsomes (ranging from 1:5 to 1:15) [46,47]. These higher CPR:CYP ratios appear to be consistent with the fact that extrahepatic organs express CYPs to a lesser extent than the liver [7,22,48].

As CYP mRNA levels are not necessarily directly correlated with CYP protein levels [49], we questioned whether CYPs activities were altered in the MCF-7/DOX^R^ cells. As such, we investigated potential deviations in drug metabolism, and the relationship between the mRNA transcripts levels and specific CYP-dependent activities in microsomes derived from MCF-7/DOX^R^ 25 nM and MCF-7/DOX^S^ cells [50,51,52] (Figure 2). The comparison of the relative velocities demonstrated that in MCF-7/DOX^R^ 25 nM cells there is: (i) no altered MROD, EROD or CECOD activities; (ii) reduced C7H activity; (iii) increased DBFOD activity. These results are indicative that at the DOX-resistance level of 25 nM, CYP2A6-dependent metabolism was significantly down-regulated, while CYP isoforms involved in DBFOD activity, particularly CYP3A4, seem to be up-regulated.

## 4. Discussion

The main cause of treatment failure in cancer is intrinsic or acquired DR, highlighting the need for a better understanding of the molecular mechanisms involved [1,3,5]. Upregulation of metabolic pathways mediated by Phase I DMEs, comprising CYP-enzyme complex protein factors and several oxidoreductases, is considered as an important potential mechanism of anticancer DR [4,7,10,13,14,19,21]. Although central in DOX metabolism, DMEs are normally underestimated in DR studies, as precedence is given to other prominent DR-associated gene families [1,5,8,19]. Together with estrogen receptors and especially drug transporters, CYPs and Phase I oxidoreductases seem to be determinant in DR. This is due to their role in drug metabolism as well as a variety of pathways that regulate cell cycle and cell growth, which are normally associated with DR mechanisms and tumor progression [23,28,29,30,31,32,33,53,54,55,56,57,58]. However, it remains unclear whether the role of the Phase I enzymes contribute to the development of early stages of chemotherapy resistance in human BC, which may enable the formation of DR to therapeutic drug levels. The novelty of our study versus a multitude of former studies on BC DR is the assessment of expression profiles of specific DMEs, as well as CYP-mediated activities, in the resistance to two sub-therapeutic DOX concentrations. Our findings helped to clarify the possible function of particular DMEs as prospective biomarkers in the emergence of DOX DR and the ability to act sooner to facilitate the adaptation to more potent and efficient therapies.

From the five genes with significantly higher levels of transcripts in both MCF-7/DOX^R^ 25 nM and 35 nM cells, *CYP26B1* was the most overexpressed, followed by *CYP8B1*, *CYP4F12*, *PAH* and *KMO*. Previous studies have shown that high expression levels of *CYP26B1* enhance the cell survival properties of breast carcinoma cells and are significantly associated with poor prognosis in colorectal cancer [28,54]. Overexpression of *CYP26B1* potentially reduces retinoic acid levels, driving cells into the oncogenic state, by altering growth, impeding differentiation, and promoting a pro-metastatic phenotype. To the best of our knowledge, differences in CYP8B1 activity have not yet been associated with any condition of drug response in BC. CYP8B1′s role in BC seems to be related to cholesterol homeostasis or molecular signaling, as it catalyzes the hydroxylation of various sterol intermediates of cholic acid in the bile acid synthesis pathway [59]. Indeed, hypercholesterolemia represents a risk factor for BC, including worse prognosis [60]. *PAH* is overexpressed in estrogen receptor-positive (ER^+^) BC patients and higher expression of *PAH* has been correlated with poor prognosis. PAH might play a role in tumor progression, as it catalyzes the rate-limiting step in the phenylalanine catabolism, converting L-phenylalanine into L-tyrosine, two essential amino acids, whose uptake and metabolism are apparently part of cancer reprogramming [53]. Integrated into the kynurenine pathway, *KMO* has been described to be upregulated in BC patients, particularly in patients with aggressive malignant BC [32,55]. It has been correlated with deregulation of genes encoding chemokines and pro-inflammatory cytokines, known to be involved in the inflammatory aspect of tumorigenesis, but also in regulation of CYP expression and other DMEs [56,61]. As such, KMO may facilitate cancer progression and chemotherapy resistance via synergistically modulating inflammatory responses in tumors with a concomitant downregulation of detoxification pathways. CYP4F12 metabolizes eicosanoids, hydroxylating arachidonic acid and its intermediate metabolite prostaglandin H2. This CYP isoform is also involved in the bioactivation of prodrugs such as ebastine and pafuramidine [11]. In a retrospective NGS study, differential activities of *CYP4F12* variants were associated with BC patients’ response to neoadjuvant cytotoxic chemotherapy [10]. *CYP4F12* expression was also correlated with tumor stage (TNM staging) [4]. This is indicative that CYP4F12 may be involved in tumor progression and drug response. Based on our data, expression levels of *CYP26B1* and *8B1* and potentially in combination with levels of *CYP4F12*, *PAH* and/or *KMO*, could act as biomarkers for the development of early stages of DR to DOX, with the practical therapeutic benefit in enabling swift changes to more effective treatment regimes, preventing full-blown DR, with obvious therapeutic advantages. Still, this needs to be verified and validated using biopsy material of BC patients undergoing DOX treatment.

Additionally, our data showed the transcription of other enzymes to be differentially regulated in DOX-resistant cells. Several epidemiological, diagnostic, and clinical studies have found that the majority of CYP genes are associated with the clinical efficacy of chemotherapy drugs in patients with BC; these genes include *CYP1A2*, *2A6*, *2D6*, *2S1*, *3A4*, and *3A5* [20,24,62,63], which were found to be deregulated in MCF-7/DOX^R^ cells, in the present study. Moreover, from the pool of differentially expressed CYP genes, five (*CYP2A7*, *2S1*, *3A5*, *4B1*, *4V2*) were previously associated with patients’ survival and suggested as potential prognosis biomarkers for several types of cancer, including BC—to evaluate tumor progression or aid decisions regarding optimal adjuvant hormonal therapy [23,29,30]. In addition, levels of flavin containing dimethylaniline monooxygenase 5 (*FMO5*) transcript, a relevant Phase I DME, were augmented in MCF-7/DOX^R^ cells. Overexpression of this monooxygenase has been associated with ERα-positive breast tumors and respective survival, and also with poor prognosis in patients with colorectal cancer [34,64]. The analysis of the mRNA levels demonstrated no significant differences in CPR transcripts between MCF-7/DOX^R^ and MCF-7/DOX^S^ cells (Figure 1A). However, CPR activity (Table 1), was significantly higher in microsomes derived from MCF-7/DOX^R^ 25 nM cells, when compared with the ones from MCF-7/DOX^S^ cells, suggesting a discrepancy between mRNA levels and phenotype. Higher levels of CPR were observed in MCF-7/DOX^R^ cells and its central key role in the metabolism of drugs, cholesterol, fatty acids, heme homeostasis (via heme oxygenase) and steroid hormone biosynthesis, is indicative of the potential implications of this oxidoreductase in tumorigenesis and cancer DR [12,21,37,65]. Other authors correlated an augmented expression of CPR in triple negative BC patients with shortened times of cancer relapse, suggesting CPR as a putative biomarker of prognosis [27].

By promoting colonization and metastasis formation, upregulation of NADH-cytochrome *b*_5_ reductase (*CYB5R*) has been previously correlated with poor prognosis in several types of cancer, including BC [31,33]. Interestingly, increased CYB5R and CPR activities have been linked to more severe thyroid neoplasms [65]. Together with arachidonate lipoxygenases (ALOXs, including *ALOXE3*), CYB5R and CPR have been suggested as critical drivers of lipid peroxidation to ferroptosis—an iron/reactive oxygen species (ROS)-dependent cell death, which plays a causative role, both in tumorigenesis progression, and in chemotherapy resistance [12]. Concomitant increased activities of these three enzymes may be related with ferroptosis and DOX’s biochemical activity, which involves the formation of ROS and potentially the reduction of iron by DOX metabolites [18]. *ALOXE3* (arachidonic acid metabolizer) and CYP4 family members (*CYP4B1*, *4F12*, *4F22*, *4V2*) are involved in the metabolism of fatty acids and fatty-acid-derived bioactive metabolites [11]. The high expression profiles of these genes observed in MCF-7/DOX^R^ cells suggest alterations in arachidonic acid metabolism and eicosanoid synthesis. Arachidonic acid metabolism upregulation has previously been linked to the induction of growth factor secretion, angiogenic factors that modulate tumor progression, and pro-inflammatory mediators, the latter of which has been associated to the regulation of DME expression [56,57,58,61].

Central in retinoic acid synthesis (β-carotene to retinaldehyde conversion), β-carotene 15,15’-monooxygenase 1 (*BCMO1*) was described previously to be involved in the modulation of migration and invasion in colorectal carcinoma cells [66]. Elevated concentrations of retinoic acid induce growth arrest, differentiation and promote cell death. However, in a feedback loop, retinoic acid, a powerful regulator of gene transcription, binds to the nuclear receptor RAR and induces the expression of *CYP26B1* (as we observed), which is involved in retinoic acid clearance [67]. Therefore, high expression levels of *CYP26B1* observed in MCF-7/DOX^R^ 25 cells could be induced by upregulation of retinoic acid metabolism via *BCMO1* overexpression, potentially driving the cells into an oncogenic state.

In addition to biogenesis of iron—sulfur clusters and heme homeostasis via ferredoxin 2 (FDX2) activity, ferredoxin reductase (FDXR) is central in sterol and vitamins synthesis, by reducing ferredoxin 1 (FDX1), the obligatory electron donor of all mitochondrial CYPs. The elevated expressions of *FDXR* and *FDX1* and the absence of CYP11 family transcripts in the MCF-7/DOX^R^ 35 nM cells is indicative of a variation in the metabolism of cholesterol into steroid hormones, which is a pathway usually deregulated in BC cells [25,26].

The measurement of specific CYP-enzyme activities suggests that CYP-dependent metabolism was altered in MCF-7/DOX^R^ 25 nM cells. Interestingly, a discrepancy was observed between the specific CYP activities and the corresponding levels of mRNA transcripts. Although no differences in the mRNA levels of CYP3A4, 3A5, 2C8, 2C9, and 2C19 were observed between MCF-7/DOX^R^ 25 nM and MCF-7/DOX^S^ cells, DBFOD activity was significantly increased in microsomes derived from the DOX-resistant cells (Figure 2). This augmented activity is coincident with a slight (although not significant) increased expression of CYP3A4 transcripts in the MCF-7/DOX^R^ 25 nM cells (not observed in the MCF-7/DOX^R^ 35 nM cells) (Figure 1A). Since CYP3A4 is described as the primary CYP isoform involved in DOX Phase I metabolism, our findings suggest that CYP3A4-dependent metabolism may be enhanced in MCF-7/DOX^R^ cells, though this may not be accompanied by a significant increase in CYP3A4 mRNA levels. C7H activity, mediated mainly by CYP2A6, decreased significantly in MCF-7/DOX^R^ 25 nM cells, contrarily to the significant 1.2 × fold change observed in the mRNA levels of the gene. Although CYP1A2 transcripts were not detected in MCF-7/DOX^R^ 25 nM cells, MROD, EROD and CECOD activities were similar to the ones determined for the naive MCF-7 cells, albeit with a slight decrease in EROD activity. The variance found in genes transcripts levels in the different types of MCF-7 cells may be a result of epigenetic modifications or copy number alterations, while the differences between gene expression and CYP activities could result from mutations or post-translational modifications—altering enzyme activity, prompted by the recognized genomic and metabolic instability of the MCF-7 cell line [68,69,70], induced by DOX exposure. Nevertheless, divergence between mRNA levels and enzyme activities is indicative that the CYP phenotype is not linearly correlated with transcription induction responses, confirming the multifactorial complexity of this mechanism [49]. By changing CYP conformation and/or catalytic turnover, specific CYP polymorphisms, post-transcriptional regulation, or distinct protein-protein interactions may contribute to this phenotypic variation [2,7,13,37,71].

## 5. Conclusions

Our findings underscore the need for additional understanding of the mechanisms involved in the early stages of DR in luminal A-like tumors, focusing on Phase I DMEs. The transcriptional analysis evidenced that acquired DR in BC cells is a dynamic process, transiently dependent on multiple pathways, including drug, cholesterol, fatty acid, and steroid metabolism beyond transport mechanisms. This suggests that mechanisms of DR may differ significantly between patients as disease progresses, adding to the complexity of underlying mechanism in the development of BC DR. In addition to evidence of deregulated drug metabolism through augmented activity of the DOX-metabolizer CYP3A4, other important pathways, previously reported to be involved in tumor progression and chemotherapy resistance, were also found to be deregulated in the first stages of acquisition of DOX-resistance in BC cells. These relate to arachidonic acid metabolism involved in the induction of growth factor secretion, angiogenic factors, and pro-inflammatory mediators, modulating tumor progression and expression of DMEs, and retinoic acid metabolism modulating cell growth and differentiation, potentially driving cells into the oncogenic state and promoting a metastatic phenotype.

## Figures and Tables

**Figure 1 genes-13-01977-f001:**
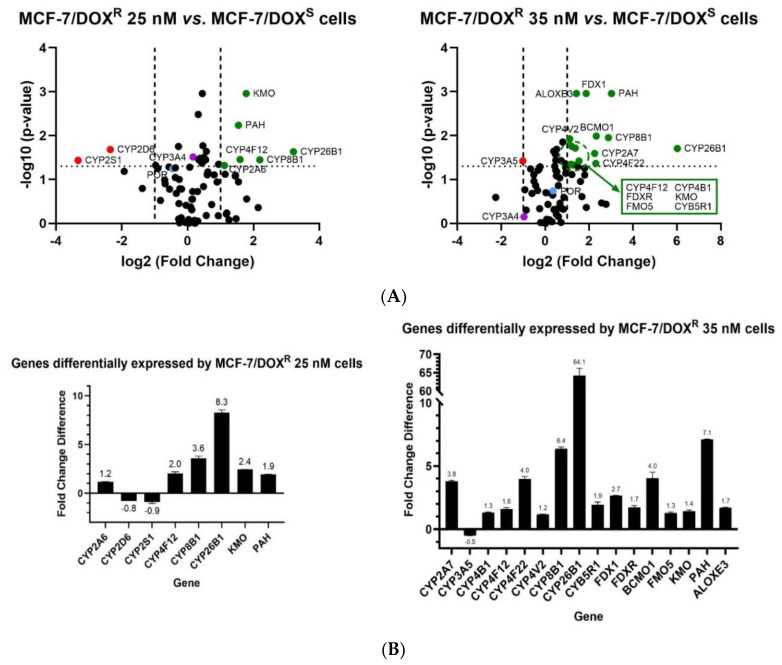
Differences of mRNA levels of target genes in MCF-7/DOX^R^ cells. (**A**) Volcano plots representing the differences in fold change of genes differentially expressed by MCF-7/DOX^R^ 25nM or 35nM cells, relative to the parental MCF-7/DOX^S^ cells. Transcripts levels were considered differentially expressed when *p* < 0.05 and fold change > 2 (upregulated, in green) or <2 (downregulated, in red). *POR* (CPR gene) and *CYP3A4* transcript levels are depicted in blue and purple, respectively. (**B**) Histograms representing the statistically significant increase (fold change) of target-gene expressions in MCF-7/DOX^R^ 25 nM or 35 nM relative to the MCF-7/DOX^S^ cells (technical replicates, N = 2). ALOXE 3: arachidonate lipoxygenase 3; BCMO1: β-carotene 15,15’-monooxygenase 1; CYB5R1: NADH-cytochrome *b*_5_ reductase 1; CYP: cytochrome P450 (isoforms); FDX1: ferredoxin 1; FDXR: ferredoxin reductase; FMO5: flavin containing dimethylaniline monooxygenase 5; KMO: kynurenine 3-hydroxylase; PAH: phenylalanine hydroxylase.

**Figure 2 genes-13-01977-f002:**
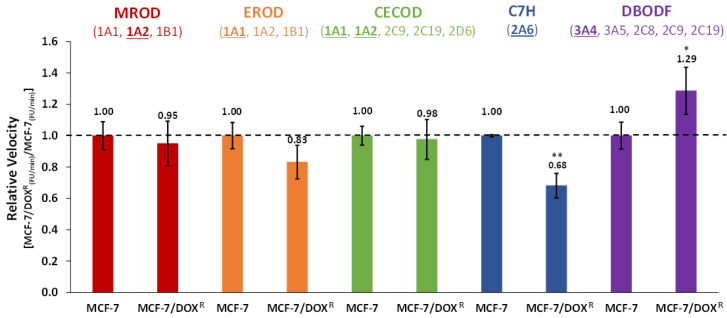
Normalized relative velocities of the CYP activity assays in MCF-7/DOX^R^ 25 nM microsomal fractions. Values are represented as mean ± SD of technical replicates (N = 3) (* *p* < 0.05; ** *p* < 0.005). MROD (red) and EROD (orange) mediated mainly by CYP1A1 (particularly EROD), 1A2 (particularly MROD), and to a minor extent by 1B1; CECOD (green) mediated particularly by CYP1A1, 1A2, and to a minor extent by 2C9, 2C19, and 2D6; C7H (blue) mediated by CYP2A6; DBFOD (purple) mediated mainly by CYP3A4, and to a minor extent by 3A5, 2C8, 2C9, and 2C19. CYP isoenzymes particularly relevant in the specific activities studied are underlined.

**Table 1 genes-13-01977-t001:** Cytochromes P450 (CYP) and cytochrome P450 oxidoreductase (CPR) contents of MCF-7-derived microsomes.

Microsomal Fractions	Protein Contents		
	CYP	CPR	CPR/CYP
	(pmol/mg Protein) ^1^		Ratios
**MCF-7/DOX^S^**	26.5 ± 5.1	15.0 ± 0.2	1:1.8
**MCF-7/DOX^R^ 25 nM**	39.2 ± 7.5	20.8 ± 0.5 *	1:1.9

^1^ CYP and CPR contents are mean ± SD (technical replicates N = 3 and N = 2, respectively). Amounts of proteins in the MCF-7/DOX^R^ 25nM-derived microsomes were compared with the ones from MCF-7/DOX^S^ cells, applying the unpaired *t* test (* *p* < 0.005).

## Data Availability

Not applicable.

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
