# Peer review of "The Complex Dynamic of Phase I Drug Metabolism in the Early Stages of Doxorubicin Resistance in Breast Cancer Cells"

_genes, 2022, doi:10.3390/genes13111977_

Round 1
Reviewer 1 Report
The study provides an experimental database with the results from expression of a set of genes, which could be used to elucidate further the mechanism of drug resistance of the chemotherapeutic doxorubicin. This is a nice summary and discussion of the existing information on DR pathways of DOX, supplemented with new data and new possible resistance-related routes. The manuscript is well written but there is one aspect that is important to address. Given the abundant research in the area (more than 4500 publications), it is necessary to highlight in a more straightforward way what is the advantage of the selected group of genes compared to previous work, e.g. of Turton et al. (10.1038/sj.onc.1204235) reporting more than 4000 genes expressed or of other authors, and what is the novelty in the current study. The latter is briefly mentioned in the conclusions but needs to be elaborated in more detail.
Author Response
The authors would like to thank the reviewer suggestion to highlight the importance of the pool of genes targeted in the study. The following sentence was added to the MS (Section 4.Discussion, lines 281-287, page 7, new version): “Although central in DOX metabolism, DMEs are normally underestimated in DR studies, as precedence is given to other prominent DR-associated gene families [1,5,8,19]. Together with estrogen receptors and especially drug transporters, CYPs and Phase I oxidoreductases seem to be determinant in DR. This is due to their role in drug metabolism as well as a variety of pathways that regulate cell cycle and cell growth, which are normally associated with DR mechanisms and tumor progression [23,28–33,53–58].” Also, we would like to substitute the last sentence of the same paragraph “To address this issue, the expression profiles of CYPs and several oxidoreductases, as well as specific CYP-mediated activities, were assessed in MCF-7 cells resistant to two sub-therapeutic DOX concentrations” by: “The novelty of our study versus a multitude of former studies on BC DR is the assessment of expression profiles of specific DMEs, as well as CYP-mediated activities, in the resistance to two sub-therapeutic DOX concentrations. Our findings helped to clarify the possible function of particular DMEs as prospective biomarkers in the emergence of DOX DR and the ability to act sooner to facilitate the adaptation to more potent and efficient therapies”. The authors added the reference 19 (Turton, N.J.; Judah, D.J.; Riley, J.; Davies, R.; Lipson, D.; Styles, J.A.; Smith, A.G.; Gant, T.W. Gene Expression and Amplifica-tion in Breast Carcinoma Cells with Intrinsic and Acquired Doxorubicin Resistance. Oncogene 2001, 20, 1300–1306, doi:10.1038/sj.onc.1204235).
Reviewer 2 Report
Authors describe the differential expression of gene implicated in drug metabolism and resistance acquisition pattern to doxorubicin in MCF7 cell model. Experimental data span among mRNA expression, protein expression and metabolic activity. The methods seem appropriate, and significant experimental data sustain the conclusions. References seem to be appropriate, in my opinion, for the scope of meaning in the discussion.
Interestingly, this paper implicitly suggests the possibility to assess drug resistance status of biopsy BC samples by assessing the expression of CYP26B1 of other key genes that are usually associated to poor prognosis in other cancers. The only limit of this paper is that there is no confirmation of these data on patients’ samples. These data, if confirmed in patients, might help in driving the choice of opportune drug therapy not substrate of the overexpressed genes in resistant phenotype, especially in initial stages of drug treatment of BC.
Minor suggestions:
1) Figure 1, please match letters A and B or (a) and (b) in figure and caption
2) Please change in appropriate font note 1 in Table 1
Author Response
The authors would like to thank the reviewer report/comments and suggestions. An additional sentence was added to the MS (Section 4.Discussion, page 8, lines 324-330, at the end of the second paragraph): “Based on our data, expression levels of CYP26B1 and 8B1 and potentially in combination with levels of CYP4F12, PAH and/or KMO, could act as biomarkers for the development of early stages of DR to DOX, with the practical therapeutic benefit in enabling swift changes to more effective treatment regimes, preventing full-blown DR, with obvious therapeutic advantages. Still, this needs to be verified and validated using biopsy material of BC patients undergoing DOX treatment.”
1) Figure 1, please match letters A and B or (a) and (b) in figure and caption. – Done.
2) Please change in appropriate font note 1 in Table 1. – Done.